# *Kazachstania pintolopesii* in Blood and Intestinal Wall of Macrophage-Depleted Mice with Cecal Ligation and Puncture, the Control of Fungi by Macrophages during Sepsis

**DOI:** 10.3390/jof9121164

**Published:** 2023-12-04

**Authors:** Pratsanee Hiengrach, Ariya Chindamporn, Asada Leelahavanichkul

**Affiliations:** 1Department of Microbiology, Faculty of Medicine, Khon Kaen University, Khon Kaen 40002, Thailand; pratsaneeh@gmail.com; 2Research and Diagnostic Center for Emerging Infectious Diseases (RCEID), Faculty of Medicine, Khon Kaen University, Khon Kaen 40002, Thailand; 3Center of Excellence in Translational Research in Inflammation and Immunology (CETRII), Department of Microbiology, Chulalongkorn University, Bangkok 10330, Thailand; 4Department of Microbiology, Faculty of Medicine, Chulalongkorn University, Bangkok 10330, Thailand; 5Mycology Unit, Department of Microbiology, Faculty of Medicine, Chulalongkorn University, Bangkok 10330, Thailand; 6Nephrology Unit, Department of Medicine, Faculty of Medicine, Chulalongkorn University, Bangkok 10330, Thailand

**Keywords:** macrophage depletion, gut dysbiosis, fungi, cecal ligation and puncture, *Kazachstania pintolopesii*, sepsis

## Abstract

Although macrophage depletion is a possible emerging therapeutic strategy for osteoporosis and melanoma, the lack of macrophage functions can lead to inappropriate microbial control, especially the regulation of intestinal microbiota. Cecal ligation and puncture (CLP) sepsis was performed in regular mice and in mice with clodronate-induced macrophage depletion. Macrophage depletion significantly increased the mortality and severity of sepsis-CLP mice, partly through the increased fecal Ascomycota, especially *Kazachstania pintolopesii*, with polymicrobialbacteremia (*Klebsiella pneumoniae*, *Enterococcus faecalis*, and *Acinetobacter radioresistens*). Indeed, macrophage depletion with sepsis facilitated gut dysbiosis that directly affected gut permeability as yeast cells were located and hidden in the colon crypts. To determine the interactions of fungal molecules on bacterial abundance, the heat-kill lysate of fungi (*K. pintolopesii* and *C. albicans*) and purified (1→3)-β-d-glucan (BG; a major component of the fungal cell wall) were incubated with bacteria that were isolated from the blood of macrophage-depleted mice. There was enhanced cytokine production of enterocytes (Caco-2) after the incubation of the lysate of *K. pintolopesii* (isolated from sepsis mice), the lysate of *C. albicans* (extracted from sepsis patients), and BG, together with bacterial lysate. These data support a possible influence of fungi in worsening sepsis severity. In conclusion, macrophage depletion enhanced *K. pintolopesii* in feces, causing the overgrowth of fecal pathogenic bacteria and inducing a gut permeability defect that additively worsened sepsis severity. Hence, the fecal fungus could be spontaneously elevated and altered in response to macrophage-depleted therapy, which might be associated with sepsis severity.

## 1. Introduction

Sepsis is an abnormal response to severe infection due to the imbalance of immune responses [1,2]. Gut dysbiosis is an imbalance in microorganisms in the gastrointestinal (GI) tract [3] that partly consists of the overgrowth of Gram-negative bacteria and fungi [4,5]. Dysbiosis in the gastrointestinal tract leads to the enhanced transfer of microbial molecules from the gut into the blood circulation (leaky gut), causing hyper-inflammatory sepsis [6]. While broad-spectrum antibiotics are important for sepsis treatment, antibiotic-induced gut dysbiosis (an increase in high-virulence organisms) might cause a more severe leaky gut that leads to gut-derived secondary infection [7], for example, antibiotic-induced candidemia [8]. Indeed, monocytes (macrophages) are important immune cells that control infections that are caused by all organisms, including fungi and bacteria [9]. Macrophages have broad-spectrum responses that can be roughly classified as classical (proinflammatory) M1 polarization and alternative (anti-inflammatory) M2 polarization [10,11]. Treatment with macrophage depletion, including bisphosphonate in osteoporosis, osteoarthritis (OA), cancer bone pain (metastases), and bone edema syndrome [12,13], is well-known with some adverse effects [14], such as nausea, vomiting, diarrhea, tiredness, confusion, and irritability [15]. Although there is no infection in the list of complications, the interference of gut microbiota and infection during treatments that cause macrophage depletion or dysfunctions are also possible. Unfortunately, data on the role of macrophages in gut microbe patterns during sepsis are still very limited.

Liposomal clodronate (dichloromethylenebisphosphonate) is a selective inhibitor of adenine nucleotide translocator (ANT; an ADP/ATP translocase enzyme) in mitochondria that induces apoptotic cell death. Liposomal clodronate, the targeted delivery of clodronate to phagocytes, is used for macrophage depletion in several models [16]. For macrophage phagocytosis, the liposome characteristics, especially the size (85 nm in diameter) and the negative electrical charge, are crucial factors for achieving a specific target within the mononuclear phagocytic system (MPS) compared with dendritic cells, neutrophils, and other lymphocytes [17,18]. Similarly, between humans and mice, liposomal clodronate inhibits the activities of osteoclasts and macrophages by suppressing cell proliferation [19,20]. Recently, Clasteon^®^ has been used in humans who have breast cancer with hypercalcemia [14]. Regarding microbial control, there might be different susceptibilities to macrophages among various organisms, for example, *Escherichia coli*, *Salmonella*, and *Candida albicans*, partly due to the resistance against macrophage phagocytosis and microbicidal activities [21]. In a previous study, macrophage-depleted mice demonstrated more severe sepsis than sepsis performed in regular mice, partly due to the enhanced leaky gut caused by gut dysbiosis from bacteria that selectively increased after macrophage depletion [22]. However, the data on (i) sepsis in conditions with macrophage depletion or hypofunction are still very low, (ii) impacts of fungemia in sepsis are still in question, and (iii) the protocol for macrophage depletion can be improved [22]. Then, we further explored the impacts of macrophage depletion on fungi during sepsis using a modified macrophage depletion protocol (daily clodronate–liposome administration) with sepsis by cecal ligation and puncture (CLP) surgery, together with several in vitro experiments.

## 2. Materials and Methods

### 2.1. Animal and Animal Model

Two-month-old C57BL/6 mice weighing 20–25 g were purchased from Nomura Siam International (Pathumwan, Bangkok, Thailand) and were used in accordance with the approved animal care and use procedures of the Institutional Animal Care and Use Committee of the Faculty of Medicine, Chulalongkorn University (035/2565), following the guidelines set by the National Institutes of Health (NIH). All mice were housed in a temperature-controlled environment (24 ± 2 °C) with 50% relative humidity and a 12 h light–dark cycle (light from 7:00 a.m. to 7:00 p.m.). Mice were maintained with a standard diet and water during the experiment. As mouse sex is associated with sepsis severity [23], only male mice were used to control this possible difference.

### 2.2. Macrophage Depletion and Sepsis Mouse Model

For macrophage depletion, clodronate–liposome (Encapsula NanoSciences LLC, Brentwood, TN, USA) with a concentration of 5 mg/mL (200 µL/mouse) was intravenously (i.v.) administered daily through the tail veins for one week, modifying from a previous report [24]. The control group was injected with liposome control. One week after injection, cecal ligation puncture (CLP) or sham surgery were performed according to an established protocol [25,26,27]. Briefly, the cecum was ligated at 1 cm from the cecal tip, punctured twice with a 21-gauge needle, and gently squeezed to release a small amount of fecal material in the peritoneum. Then, tramadol (25 mg/kg/dose) in 0.25 mL prewarmed normal saline solution (NSS) was subcutaneously administered in abdominal areas at 6 h and 18 h after surgery. In sham-operated mice, the cecum was isolated and the abdominal wall was closed by suturing without ligation or puncture. Mice were sacrificed with sample collection by cardiac puncture under isoflurane anesthesia. Blood samples were centrifuged to remove the cells and serum before being kept at −80 °C until analysis. Organs were fixed with 10% paraformaldehyde or Cryogel (Leica Biosystems, Richmond, IL, USA) for histological analysis and immunofluorescent staining, respectively.

### 2.3. Detection of Bacteremia and Fungemia

To investigate microorganism translocations, both bacteremia and fungemia in macrophage-depleted mice and fresh blood (100 μL) with serial dilutions were plated onto both blood agar (HiMedia; Mumbai, Maharashtra, India) and Sabouraud dextrose agar (SDA; Oxoid, Hampshire, UK). All plates were incubated at 35 °C overnight before performing colony enumerations. The colonies from culture plates were extracted for DNA using phenol:chloroform as previously described [28]. The amplification of 16S rRNA and internal transcribed spacer (ITS) genes was used to identify bacterial species and fungal species, respectively. The BLAST-based tool was used (https://blast.ncbi.nlm.nih.gov (accessed on 4 April 2023) [28].

### 2.4. Histological Analysis

To determine macrophages in the tissue, immunohistochemistry staining using antibodies against F4/80 (ab6640, Abcam, Cambridge, UK) and arginase-1 (Arg1) (GeneTex, GTX113131, Irvine, CA, USA) was used with a goat anti-rabbit IgG secondary antibody (Alexa Fluor^®^ 488, ab150077) before visualization using a ZEISS LSM 800 confocal microscope (Zeiss, Jena, Germany). In parallel, the internal organs (livers, kidneys, and colons) were preserved in 10% paraformaldehyde, embedded in paraffin, and sectioned at 5 µM thickness before staining with Hematoxylin and eosin color (H&E; Sigma-Aldrich, St. Louis, Missouri, USA) according to the standard protocol [29,30]. Then, the slides were analyzed in 10 randomly selected areas. Kidney injury scores were determined by tubular cell damage with the scores as previously published [31,32], including, 0; no damage, 1; 0–25% tubular damage, 2; 25–50% tubular damage, 3; 50–75% tubular damage, and 4; 75–100% tubular damage. Liver damage was determined based on inflammation and hepatocyte necrosis with the following scores, including 0 (no finding); 1 (mild); 2 (moderate); and 3 (severe) [25,33,34,35]. Colon damage scores were evaluated using the Geboes Score (GS) based on the erosions or ulcerations, mononuclear cell infiltration, and structural alterations with scores ranging from 0–5 [36]. Additionally, Gomeri Methenamine Silver (GMS) color (Sigma-Aldrich) was used to identify fungi in the colon. Moreover, the fungi in the colon were also identified by culture and polymerase chain reaction.

For tissue cytokines, the organs (kidneys and livers) were homogenized in ice-cold phosphate buffer solution (1xPBS, pH 7.4) with 45 s sonication using pulse-on and pulse-off for 20 and 5 s, respectively, (Sonics Vibra Cell, VCX 750, Sonics & Materials Inc., Newtown, CT, USA) before the separation of the supernatant with centrifugation [37]. Then, cytokines (IL-6, TNF-α, and IL-10) were measured using the Enzyme-linked immunosorbent assay (ELISA) (Invitrogen, Waltham, MA, USA).

### 2.5. Mouse Serum Analysis and Intestinal Permeability Test (Leaky Gut)

QuantiChrom Creatinine and EnzyChrom ALT assays (BioAssay, Hayward, CA, USA) were used to determine creatinine and alanine transaminase (ALT) in serum, respectively. Serum cytokines were evaluated by ELISA (Invitrogen). To detect leaky gut, fluorescein isothiocyanate-dextran (FitC-dextran) (25 mg/mL) was orally administered to mice, and mouse blood was collected 3 h later by tail vein nicking before quantification of the blood FitC-dextran levels. The serially diluted FitC-dextran was used as the standard curve to determine FitC-dextran values using a fluorospectrometric machine with excitation and emission wavelengths at 485 and 528 nm, respectively. Owing to the possible sustaining of FitC-dextran in the gut for a few days after an oral administration [38,39,40,41], different mice were used for the time-point experiments.

### 2.6. The Fungal Burden in Colon Tissue

Colon samples (0.1 g) were gently ground and homogenized in 1xPBS (pH 7.4). Subsequently, 100 μL of each homogenized sample was spread onto SDA supplemented with chloramphenicol, and the plates were incubated at 35 °C for 48 h before colony enumerations. Furthermore, fungal abundance within the colon tissue was determined through the quantitative polymerase chain reaction (qPCR) using 0.1 g of the colon to extract DNA with phenol:chloroform methods. The fungal-specific primers ITS1 (forward: 5′-TTCGTAGGTGAACCTGCGG-3′ and ITS4 reverse; 5′-TCCTCCGCTTATTGATATGC-3′) [42,43] were used to identify species using the BLAST-based tool within the NCBI GenBank database (https://blast.ncbi.nlm.nih.gov (accessed on 4 April 2023)) [28].

### 2.7. Gut Mycobiome Analysis

An amount of 0.3 g of mouse feces was collected for gut microbiome analysis following a previous report [44]. In brief, fecal DNA was extracted with the phenol:chloroform method. The purified metagenomic DNA was used as the template for the ITS region and ITS genes were sequenced using the Illumina Miseq sequencing platform (Illumina, San Diego, CA, USA) with the fungal-specific primers ITS1 (forward: 5′-TTCGTAGGTGAACCTGCGG-3′ and ITS4 reverse; 5′-TCCTCCGCTTATTGATATGC-3′) for gut mycobiome identification [42,43]. The raw sequences and operational taxonomic units (OTUs) of the DNA library were sequenced using the Miseq system (Illumina) at Omics Sciences and Bioinformatics Center, Chulalongkorn University. Forward and reverse primers were removed from raw sequences using cutadapt v 1.18. and trimmomatic v 0.39 with the sliding window option to trim individual sequences where the average quality scores were less than 15 across 4 base pairs. The amplicon sequence variants (ASVs) were analyzed by the QIIME2 plugin DADA2 pipeline software for the identification of the composition of fungi in fecal samples without unclassified phylum or higher fungal classification. The fungal classification was analyzed using the BLAST-based tool (https://blast.ncbi.nlm.nih.gov/, (accessed on 29 April 2023)) [44].

### 2.8. The In Vitro Experiments

The impacts of the molecular components of fungi and bacteria were explored. As such, the organisms that were isolated from the blood of clodronate-administered mice, including bacteria (*K. pneumoniae*, *E. faecalis*, and *A. radioresistens*) and a fungus (*K. pintolopesii*), were inoculated into Tryptic Soy Broth (TSB; HiMedia, Mumbai, Maharashtra, India) at 37 °C and Sabouraud dextrose broth (SDB; Oxoid) at 35 °C overnight, respectively. The organisms were heat-killed at 60 °C for 1 h with sonication for fungal lysate and bacterial lysate. To explore the possible impact of fungal molecular components on bacterial growth, the isolated bacteria (listed above) at 1 × 10^9^ cells/ mL were incubated with the heat-killed fungal lysate (0.1 mL) using a multiplicity of infection (MOI) of fungi versus bacteria at 1, 0.1, and 0.01 on Tryptic Soy Agar (TSA) plates for 24 h before colony enumeration. In parallel, (1,3)-β-D-glucans (BG) (Pachyman; megazyme, Ireland) was also incubated with bacteria to test the possible influence of BG, a major cell wall component of fungi, on bacterial growth. As such, the isolated bacteria (listed above) at 1 × 10^9^ cells/mL were incubated with BG (1 mg) on TSA plates for 24 h before colony enumeration. Additionally, to test the microbial impact on enterocytes, the colorectal cell line (Caco-2) (ATCC HTB-37, USA) was cultured in Dulbecco’s Modified Eagle Medium (DMEM) supplemented with 20% heat-inactivated fetal bovine serum (FBS; Gibco, Carlsbad, CA, USA) and 1% PenStrep at 37 °C in a 5% CO_2_ incubator. Then, Caco-2 (1 × 10^6^ cells/ well) was incubated with fungi lysate (MOI 0.1), bacteria lysate (MOI 0.01), the mixed lysate (fungi and bacteria), or BG (1 mg) for 24 h at 37 °C with 5% CO_2_ before measuring supernatant IL-8 using ELISA (Invitrogen).

### 2.9. Statistical Analysis

Data were presented as the mean ± standard error (SE) using SPSS 11.5 and GraphPad Prism version 7.0 for statistical analysis. One-way analysis of variance (ANOVA) followed by Tukey’s analysis and Student’s *t*-test was used for multiple-group and two-group comparisons, respectively. A *p*-value of less than 0.05 was considered statistically significant.

## 3. Results

### 3.1. Spontaneous Bacteremia and Fungemia after Macrophage Depletion without Sepsis, the Leaky Gut-Induced Systemic Inflammation from the Loss of Macrophage Microbial Control

The clodronate–liposome (Clod) protocol (Figure 1A) effectively depleted macrophages as indicated by the absence of the F4/80-positive cells and arginase 1 (Arg-1)-positive cells when compared with the blank liposome (Lipo) control mice (Figure 1B–F).

In comparison with Lipo control, one week of macrophage depletion (without CLP surgery) induced a more severe condition as indicated by mortality, weight loss, leaky gut, serum cytokines (TNF-α and IL-10, but not IL-6), and organ injury (histological scores and organ cytokines of liver, kidney, and colon), but neither serum creatinine nor alanine transaminase (Figure 2A–H and Figure 3A–L).

Indeed, enhanced neutrophil infiltration into the organs of Clod-administered mice (liver, kidney, and colon) compared with the control was also demonstrated (Figure 4A–F).

Surprisingly, spontaneous bacteremia and fungemia in macrophage-depleted mice without sepsis induction were identified from the microbial colonies in the culture plates as *Kazachstania pintolopesii* and *Acinetobacter radioresistens*, respectively, by mass spectrometry analysis (see Section 2) (Figure 5A–C).

Additionally, enhanced fungi in the feces of macrophage-depleted mice were also demonstrated by fecal mycobiome analysis. The ITS sequence reads were processed using the DADA2 pipeline software, indicating a comprehensive total frequency of 766,274 reads, with an average of 63,857 reads per sample. The mycobiome analysis demonstrated an increased abundance of Ascomycota (the phylum of *Kazachstania* spp.) with increased evenness estimation (Shannon score) (Figure 6A–L). Moreover, the non-metric multidimensional scaling function (the difference among experimental groups) based on the J class index (community membership) indicated possible differences in the fecal fungi among macrophage-depleted mice and other experimental groups (Figure 6J–L).

However, the fungal abundance of macrophage-depleted mice in colon tissue was not high enough to detect the fungi by Gomeri Methenamine Silver (GMS) color staining (Figure 7A–D).

### 3.2. More Severe Sepsis in Macrophage-Depleted Mice with Cecal Ligation and Puncture Surgery, a Possible Impact of Leaky Gut and Fungi in the Intestinal Tissue

Cecal ligation and puncture (CLP) surgery was performed seven days after the administration of clodronate (Clod) or liposomal control (Lipo) (Figure 1A). As such, there was an enhanced severity of sepsis in macrophage-depleted mice with CLP compared with CLP in the Lipo group, as indicated by the survival analysis, leaky gut, serum cytokines (TNF-α and IL-10, but not IL-6), serum creatinine, and alanine transaminase (Figure 2A–H). In parallel, the histological score of colons (neither liver nor kidney), cytokines from the liver tissue (TNF-α and IL-6, but not IL-10), and cytokines from kidney samples (TNF-α, IL-6, and IL-10), inflammatory cell infiltration of the internal organs (livers, kidneys, and colons), and fungemia (but not bacteremia) in macrophage-depleted CLP mice were also higher than those in Liposome-CLP mice (Figure 3A–L, Figure 4A–F and Figure 5A–C). Hence, there was a two-hit injury in macrophage-depleted mice with CLP as clodronate administration firstly induced systemic inflammation and CLP surgery secondly worsened the injury later. Surprisingly, some of the parameters of macrophage-depleted mice (Clod-sham) were as severe as those of the macrophage-depleted CLP group (Clod-CLP), including kidney injury scores (Figure 3B), neutrophil infiltrations in kidneys and colons (Figure 4B,C), and fungemia (Figure 5B). Most injury parameters in Clod-CLP mice were more prominent than those in Clod-sham mice. Notably, the levels of bacteremia, but not fungemia, in Clod-CLP mice were higher than those in the Lipo-CLP group (Figure 5A,C). In parallel, the identified bacteria from the colonies in culture plates were *K. pneumoniae*, *E. faecalis*, and *A. radioresistens*, whereas fungi from the plates were *K. pintolopesii* (Figure 5B,C). Macrophage depletion may interfere with the microbial control mechanisms in the gut and lead to increased fungal abundance, which escalates leaky gut (the translocation of microbial molecules from the gut into the blood circulation).

There was a subtle change in fecal mycobiome analysis among the experimental groups (Figure 6A–D). As such, the fecal fungi were mostly in the phylum Ascomycota, especially in macrophage-depleted mice without sepsis (Clod-sham) and the intact macrophages with sepsis (Lipo-CLP), while macrophage-depleted sepsis (Clod-CLP) and control mice (Lipo-sham) demonstrated lower abundance (Figure 6E). Only a few fungi in genus level were different among these groups (Figure 4D), including *Debaryomyces* spp., which was lowest in Clod-CLP, while *Myrothecium* spp. and *Cladosporium* spp. were not different among groups (Figure 6F–H). The alpha diversity, as indicated by Shannon evenness estimation in Clod-sham, was higher than those in Clod-CLP and Lipo-sham (Figure 6I). Meanwhile, the beta diversity using Bray–Curtis analysis (determining the difference among groups by distances from the axis) of only Clod-sham seemed to be different from that of the other groups (Figure 6J). Notably, the abundance of fecal fungi as determined by fecal operational taxonomic units (OTUs; clusters of the microorganisms grouped by their DNA sequence similarity) was similar among all groups (Figure 6K). Here, over 90% of reads in 8 of the 12 samples were classified as ‘unidentified fungi’, indicating the limitation of fungal identification tools; thus, further studies might be essential to address this limitation comprehensively with more effective fungal analysis methodologies. Interestingly, the subtle difference in the fecal mycobiome might partly be due to the adherence of the fungi to the intestinal wall [45,46]. Indeed, the Gomeri Methenamine Silver (GMS) staining was able to identify fungal cells attached to the colon tissue (Figure 7A–D). Although the black color of GMS was detectable in the colon cell wall of all groups, the GMS-stained fungi, as indicated by the yeast-like structural morphology, were detectable only in macrophage-depleted mice with sepsis (Figure 7A–D). These data also supported the vulnerability of the macrophage-depleted intestine during sepsis to enhanced fungal adherence onto the intestinal tissue.

### 3.3. The Overgrowth of Bacteria due to the Fungi and the Mixed Microbial Activation of Enterocytes, the Possible Influence of Gut Fungi on Sepsis Severity

Due to (i) the co-identification of bacteria (*K. pneumoniae*, *E. faecalis*, and *A. radioresistens*) and fungi (*K. pintolopesii*) in macrophage-depleted mice (Figure 5B,C) and (ii) the selection of bacteria by some fungi [22], the correlation or interactions among these organisms was possible. Although there are several possible hypotheses on bacterial–fungal interactions, fungal molecules in the cell wall components of *K. pintolopesii* might induce the overgrowth of these specific bacteria. To demonstrate the possible impacts, the heat-kill lysate of *K. pintolopesii* (K. pin) isolated from our mice and the heat-kill lysate of *C. albicans* (C. al) isolated from the blood of sepsis patients were incubated with these isolated bacteria, including *K. pneumoniae* (Kleb), *E. faecalis* (Entero), and *A. radioresistens* (Acineto). Interestingly, the lysate of both isolated yeast cells (*C. albicans* and *K. pintolopesii*) significantly increased bacterial overgrowth in a dose-dependent manner, especially the co-incubation at the multiplicity of infection (MOI) at 0.01 of bacteria with the lysate of both isolated yeast cells when compared with the non-fungal lysate (Figure 8A–L).

As β-d-glucan (BG) is a major component of the fungal cell wall that could promote the growth of some bacteria species [22,47], BG from the gut fungi that are elevated after macrophage depletion might be responsible for the enhanced growth of some bacteria, especially the bacteria isolated from macrophage-depleted mice. Indeed, the purified BG in different doses (0.1 mg and 1 mg) facilitated the growth of these bacteria, isolated from macrophage-depleted mice, in a dose-dependent manner when compared with the non-BG groups (Figure 9A–C). As the fungi (*K. pintolopesii* in mice or *Candida* spp. From humans) and purified BG might not only enhance bacterial abundance, but also activate enterocytes, Caco-2 cells (enterocytes) were incubated with the lysate of fungi (K. pin or C. al) or purified BG with or without bacterial lysate (Kleb, Entero or Acineto). Supernatant IL-8, a cytokine highly produced by Caco-2, was used as an indicator of enterocytic responses [48]. As such, the additive effect on supernatant IL-8 between the lysate of bacteria or fungi alone versus the bacteria plus fungi was demonstrated only with the use of Acineto and Kleb plus fungal lysate (K. pin or C. al), but not the Entero with fungi (Figure 9A,B). These effects were partly explained through the additive enterocyte responses against bacteria and BG of fungi as Acineto plus fungi induced higher supernatant IL-8 than Acineto alone (Figure 9C). However, there was no additive effect by BG together with Kleb or Entero (Figure 9C), perhaps due to the necessity of a proper ratio between BG and bacterial molecules. More studies would be interesting.

## 4. Discussion

### 4.1. Macrophage Depletion Facilitated the Growth of Gut Fungi

Macrophages, the well-known sentinel immune cells in all organs, are peripheral blood monocytes that migrate into several organs [49,50,51]. With daily clodronate administration [52,53], the depletion of macrophages was supported by the immunofluorescent pictures of the livers and colons. Although macrophage depletion [54] was achieved by approximately 80–90% within 24–48 h after a single intravenous (i.v.) or intraperitoneal (i.p.) injection in a previous report [55], the repeated doses of clodronate would increase the effectiveness of macrophage depletion [24]. Despite the possible more complete macrophage depletion in our current model, the increased severity in the CLP sepsis model was similar to our previous model with the less frequent clodronate injection [22]. Here, macrophage depletion alone caused gut permeability defect (gut leakage) and gut mycobiome imbalance (increased fecal fungi). Accordingly, *Candida pintolopesii*, scientifically designated as *Kazachstania pintolopesii*, is an integral constituent of the genus *Kazachstania* and forms an essential part of the *Kazachstania telluris* species complex, which comprises a total of five distinct yeast species. This specific microorganism holds significant scientific relevance within the field of microbiology and contributes to the broader understanding of fungal diversity and classification [56]. Indeed, *K. pintolopesiis* is a fungus frequently found in the mouse intestinal mucosa [57] and is associated with the upregulation of IL-17 receptor A (IL-17RA) and IL-23, resulting in a severe inflammatory reaction in mouse colons [58]. In this study, *K. pintolopesiis* was identified from the blood of sepsis mice with or without clodronate (more profound in the clodronate group), supporting the translocation of the fungi from the gut of macrophage-depleted mice as previously mentioned [22]. While it is well-known that sepsis causes enterocyte damage and leaky gut through several mechanisms (hypoperfusion and inflammatory responses) [2,59], data on enhanced fungal translocation from the gut during bacterial sepsis are still lacking [60]. Despite the lack of information on fungi in the gut during bacterial sepsis, increased fecal fungi in chronic intestinal inflammation from inflammatory bowel disease (IBD) is well-established [61]. In IBD, increased fungal abundance in the gut is partly explained by inflammation-induced intestinal barrier defects, resulting in the reduction of bacteria that possess fungal control activities and/or the facilitation of fungal attachment onto gut mucosa [28,61]. In our study, macrophage depletion increased fungal overgrowth in sepsis, as indicated by detecting fungi in colon tissue using PCR, culture, and GMS staining (Figure 7). Hence, the discordance of fungal abundance from the samples of feces alone versus feces with intestinal walls due to fungal adherence to the intestinal mucosa is possible. While *C. albicans* is common in the human gut [62,63,64] and *C. tropicalis* is associated with IBD in patients [61], the isolated *K. pintolopesii*, the common fungi in laboratory mice [57,65] might be associated with mouse sepsis severity. As macrophages play a crucial role in the maintenance of intestinal homeostasis, especially gut fungi [22], the fungal overgrowth in the condition without macrophages is not surprising. Although *Cladosporium* spp., an environmental saprobic fungus causing infections in some conditions [66,67], was also increased in Clod-CLP mouse feces, the culture could not identify the fungus. Hence, we further evaluated the possible impact of *K. pintolopesii*, but not the *Cladosporium* group.

### 4.2. K. pintolopesii as a Source of Fungal Molecules Influenced Intestinal Bacteria and Enterocytes

The interaction between fungi and bacteria is well-known through several mechanisms. For example, the fermentation of (1→3)-β-d-Glucan (BG) on the fungal cell wall by bacterial endo-β-glucanase enzyme [68] causes an overgrowth of pathogenic bacteria [47], resulting in dysbiosis in several conditions [69,70].

As BG is the major component of *K. pintolopesii* isolated from our models, BG from these fungi might, at least in part, induce the growth of the co-existing bacteria (*Klebsiella* spp., *Enterobacter* spp., and *Acinetobacter* spp.), leading to the positive culture of these organisms in our model (Figure 10). Indeed, the lysate of *K. pintolopesii* facilitated the growth of these isolated bacteria, similar to the use of *C. albicans* lysate and purified BG, supporting that BG in the fungal cell wall is partly responsible for the overgrowth of pathogenic bacteria and possibly leading to gut dysbiosis. Likewise, gut dysbiosis and leaky gut in mice orally administered with heat-killed *C. albicans* have previously been mentioned [31,71,72]. The additive pro-inflammation of BG upon LPS responses has been demonstrated in many mouse models [6,73] partly because of the co-presence of Dectin-1 and TLR-4, the receptors for BG and LPS, respectively [31,71], that synergistically elevate cytokine production [74]. Hence, macrophage dysfunction possibly enhances gut fungi, especially *K. pintolopesii*, causing the overgrowth of pathogenic bacteria, leading to a more severe leaky gut that worsens sepsis severity. It is well-known that enterocytes have resistance against LPS, as a high dose of LPS is necessary for inducing enterocytic inflammatory responses [75,76]. However, the combination of heat-killed lysate of Gram-negative bacteria (*Acinetobacter* spp. and *Klebsiella* spp.) and the fungal lysate elevated enterocyte IL-8 production (Caco-2 cells) (Figure 9A–C) supported an additive inflammatory effect with the co-presence of fungi together with bacteria. Nevertheless, there was no additive pro-inflammatory effect on fungi plus *Enterobacter* spp., and only BG plus *Acinetobacter* lysate (not the lysate from other bacteria) enhanced enterocyte IL-8 production (Figure 9A–C). These data imply a possible ratio of LPS and BG for maximizing the enhanced inflammatory activation on enterocytes. More studies would be interesting.

### 4.3. The Discrepancy between Fecal Mycobiome Analysis versus Fecal Fungi from Polymerase Chain Reaction (PCR), a Possible Role of the Internal Transcribed Spacer (ITS) Primers

Here, there was a fundamental discrepancy between the mycobiome results (Figure 6) that neither Saccharomycetaceae (Family) nor Kazachstania (Genus) were detectable, while the fecal *K. pintolopesii* could be detected by PCR. This might be a result of the use of the entire ITS region (ITS1 and ITS4) for the high-throughput profiling of the fungal mycobiome in our study. ITS is the spacer DNA between the small-subunit ribosomal RNA (rRNA) and large-subunit rRNA or the corresponding transcribed region in the polycistronic rRNA precursor transcript [77]. There is only a single ITS between the 16S and 23S rRNA genes in bacteria and archaea, while there are several ITSs in eukaryotes, including fungi. For instance, ITS1 is located between the 18S and 5.8S rRNA genes, while ITS2 is between the 5.8S and 28S (or 25S in plants) rRNA genes. Interestingly, the utilization of different ITS primers provides different results in mycobiome analysis [78]. For example, the primers ITS1-F and ITS1 are biased towards the amplification of basidiomycetes, whereas ITS2, ITS3, and ITS4 are biased towards the positive results on ascomycetes. Meanwhile, the basidiomycete-specific primer (ITS4-B) can only amplify a minor proportion of basidiomycete ITS sequences [79]. The entire ITS region might be suitable for performing species identification by colony PCR (using conventional Sanger sequencing), but it might not be compatible with the high-throughput Illumina MiSeq sequencing due to the read length limitation of the MiSeq Illumina sequencing (150–300 base pairs). Hence, either the ITS1 or ITS sub-region might be more suitable to select for fungal community profiling [80]. Indeed, the high throughput using ITS1 MiSeq sequencing can identify *K. pintolopesii* as a key member of the captive cynomolgus macaque intestinal mycobiota [65]. Despite these technical limitations, our data demonstrated some possible impacts of macrophages on the fecal fungal population through the difference between clodronate-administered mice versus the control intact macrophage group. Although the technical limitations might have influenced the accuracy of the lists of the discovered fungi, a difference in fecal fungi between the macrophage-depleted group and intact macrophage mice was demonstrated. Further studies on the mycobiome with the proper use of ITS primers would be interesting.

## 5. Conclusions

In conclusion, macrophage depletion alone (using daily liposomal clodronate) without sepsis altered the fungal component in mouse feces (mycobiome analysis) without fungal detection in the intestinal wall (Figure 7). However, sepsis induction using CLP in macrophage-depleted mice induced the presence of fungi in the intestinal wall with a subtle change in fecal mycobiome analysis when compared with macrophage-depleted mice without sepsis. The increased fungi facilitated the growth of some bacteria, causing gut dysbiosis through enhanced gut inflammation by molecules from fungi plus bacteria that worsened leaky gut, bacteremia, and fungemia, leading to more severe sepsis. Hence, sepsis during some treatments that deplete macrophages should be monitored. More studies are warranted.

## Figures and Tables

**Figure 1 jof-09-01164-f001:**
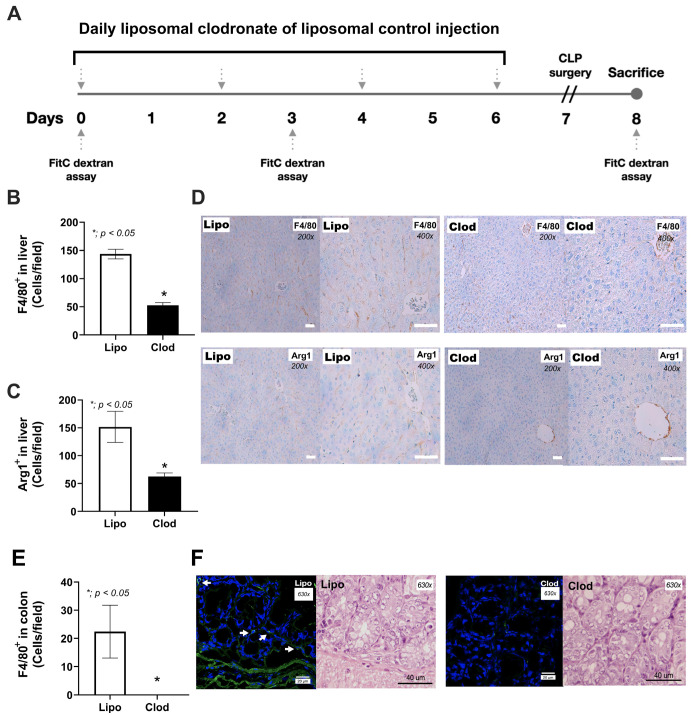
The schema of experiment (**A**) using liposomal clodronate (Clod) or liposomal control (Lipo) injection and the determination of leaky gut (FitC-dextran assay) are shown. The representative immunohistochemistry pictures (anti-F4/80 and anti-arginase 1) in the liver with the scoring analysis (**B**–**D**) are presented (the white bars in (**D**) represent 40 µm) (n = 5/group). The immunofluorescent pictures of anti-F4/80 in the colon with the score (**E**,**F**) are also provided (n = 5/group). The data are shown as the mean ± SE, *; *p* < 0.05 vs. Lipo using Student’s *t-*test analysis. White arrows indicate F4/80-positive cells in the colon and the brown color-stained cells represent F4/80 and/or arginase 1-positive cells in the liver.

**Figure 2 jof-09-01164-f002:**
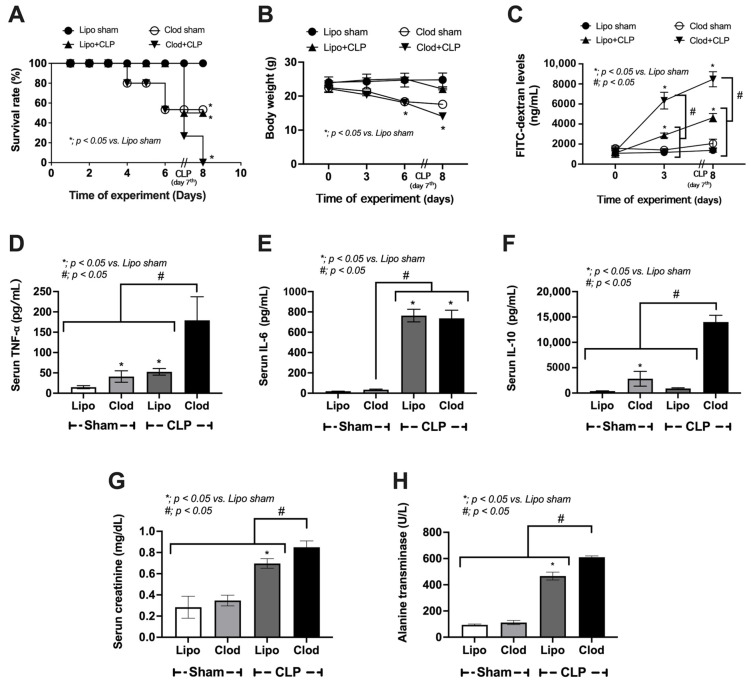
Characteristics of mice with sham surgery and liposome control (Lipo-sham) or clodronate–liposome (Clod-sham) or cecal ligation and puncture (CLP) sepsis surgery with liposome control (Lipo-CLP) or clodronate–liposome (Clod-CLP) as indicated by survival analysis (**A**) with the time-point parameters of body weight (**B**) and FitC-dextran levels (**C**), together with several parameters at 24 h post-surgery, including serum cytokines (TNF-α, IL-6, and IL-10) (**D**–**F**) and kidney damage (serum creatinine) (**G**), and liver injury (serum alanine transaminase) (**H**) are demonstrated, (n = 10/group for (**A**) and n = 5–7/group for (**B**–**H**)). The data are shown as the mean ± SE, *; *p* < 0.05 vs. Lipo-sham, and #; *p* < 0.05 between the indicated groups using ANOVA with Tukey’s analysis.

**Figure 3 jof-09-01164-f003:**
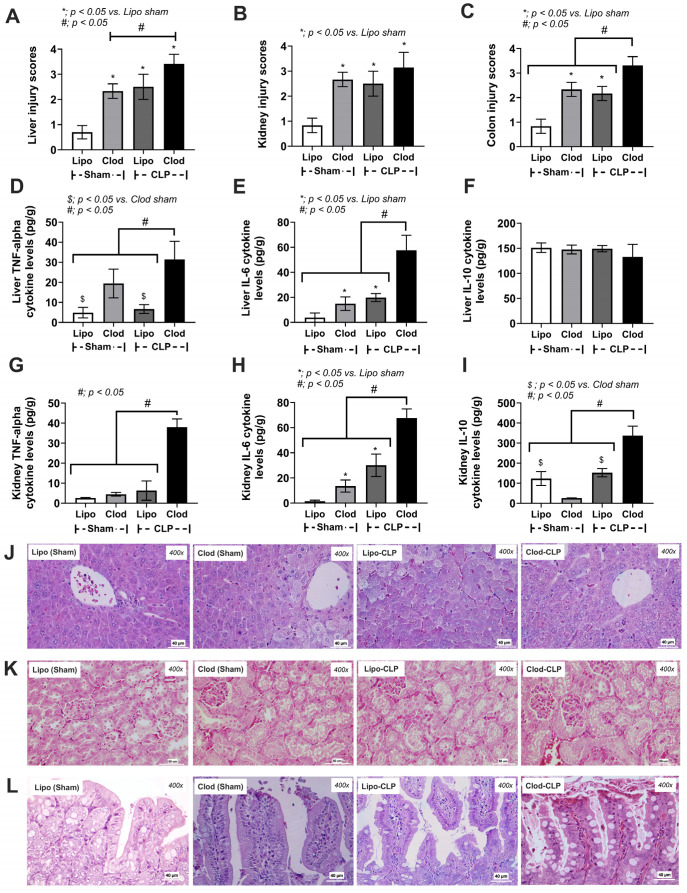
Characteristics of mice with sham surgery and liposome control (Lipo-sham) or clodronate–liposome (Clod-sham) or cecal ligation and puncture (CLP) sepsis with liposome control (Lipo-CLP) or clodronate–liposome (Clod-CLP) as indicated by organ injury scores (liver, kidney, and colon) (**A**–**C**), cytokines (TNF-α, IL-6, and IL-10) in liver and kidneys (**D**–**I**), and the representative histological pictures (H&E staining) of these organs (liver, kidney, and colon (**J**–**L**) (n = 5/group). The data are shown as the mean ± SE, *; *p* < 0.05 vs. Lipo-sham, $; *p* < 0.05 vs. Clod-sham and #; *p* < 0.05 between the indicated groups using ANOVA with Tukey’s analysis.

**Figure 4 jof-09-01164-f004:**
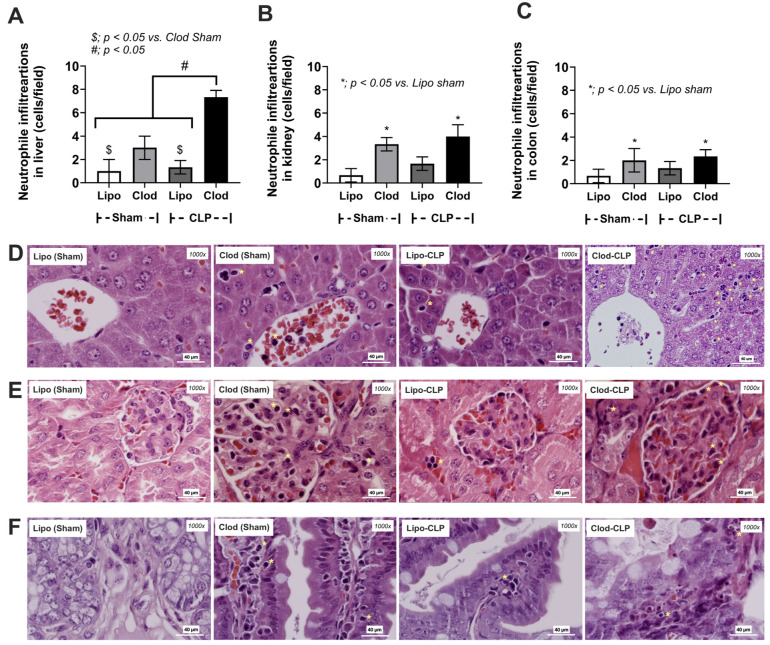
Characteristics of mice with sham surgery and liposome control (Lipo-sham) or clodronate–liposome (Clod-sham) or cecal ligation and puncture (CLP) sepsis with liposome control (Lipo-CLP) or clodronate–liposome (Clod-CLP) as indicated by neutrophil infiltration, determined by cell morphology in the H&E staining, especially polymorphonuclear cells, in several organs (liver, kidney, and colon) with the score analysis (**A**–**C**) and the representative histological pictures (**D**–**F**) (n = 5/group). The data are shown as the mean ± SE, *; *p* < 0.05 vs. Lipo-sham, $; *p* < 0.05 vs. Clod-sham, and #; *p* < 0.05 between the indicated groups using ANOVA with Tukey’s analysis. The yellow star indicates neutrophils in the organs.

**Figure 5 jof-09-01164-f005:**
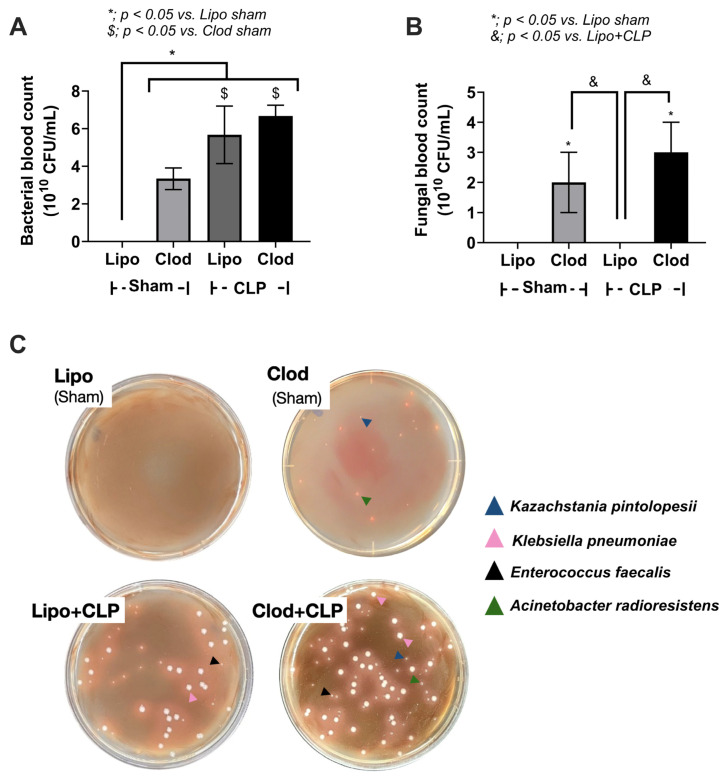
Characteristics of mice with sham surgery and liposome control (Lipo-sham) or clodronate–liposome (Clod-sham) or cecal ligation and puncture (CLP) sepsis with liposome control (Lipo-CLP) or clodronate–liposome (Clod-CLP) as indicated by the enumeration of bacteria and fungi in blood (**A**,**B**) with the representative colonies from the culture plates, Sabouraud dextrose agar (SDA) (**C**) (n = 5/group). The data are shown as the mean ± SE; *, *p* < 0.05 vs. Lipo-sham; $, *p* < 0.05 vs. Clod-sham; and &, *p* < 0.05 vs. Lipo+CLP group using ANOVA with Tukey’s analysis. The colonies were identified by mass spectrometry analysis (see method) as *Kazachstania pintolopesii* (navy triangle), *Klebsiella pneumoniae* (pink triangle), *Enterococcus faecalis* (black triangle), and *Acinetobacter radioresistens* (green triangle).

**Figure 6 jof-09-01164-f006:**
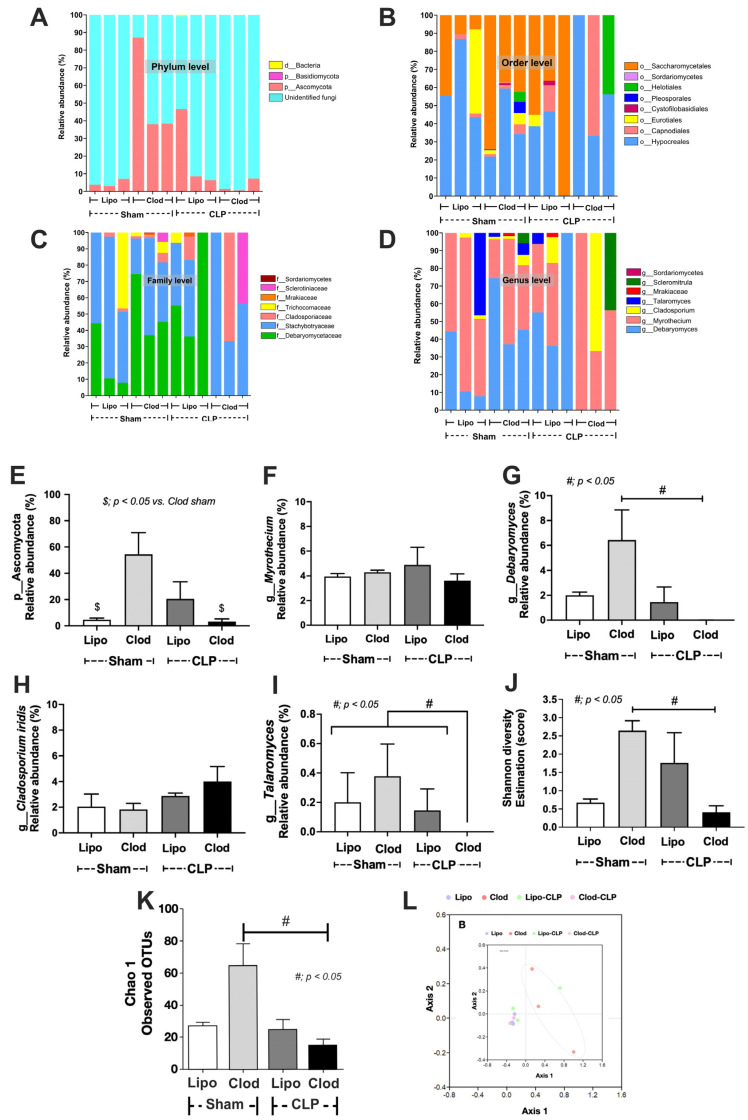
Fungal microbiome (mycobiome) analysis from feces of mice with sham surgery and liposome control (Lipo-sham) or clodronate–liposome (Clod-sham) or cecal ligation and puncture (CLP) sepsis with liposome control (Lipo-CLP) or clodronate–liposome (Clod-CLP) as indicated by the relative abundance at the phylum, order, family, and genus levels (**A**–**D**), graph presentation of the relative abundance of some fungal groups (**E**–**I**), alpha diversity (Shannon and Chao-1) (**J,K**), and beta diversity (Bray-Curtis) (**L**). The data are shown as the mean ± SE, $; *p* < 0.05 vs. Clod-sham group and #; *p* < 0.05 between the indicated groups using ANOVA with Tukey’s analysis.

**Figure 7 jof-09-01164-f007:**
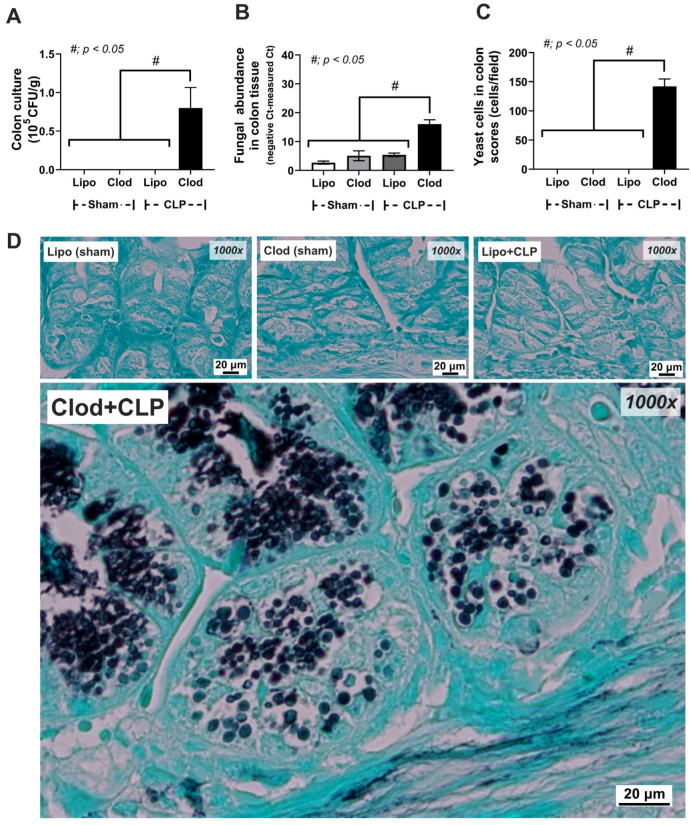
Identification of fungi from colon tissue of mice with sham surgery and liposome control (Lipo-sham) or clodronate–liposome (Clod-sham) or cecal ligation and puncture (CLP) sepsis with liposome control (Lipo-CLP) or clodronate–liposome (Clod-CLP) as indicated by culture (**A**), polymerase chain reaction (PCR) (**B**), and Gomeri Methenamine Silver (GMS)-stained slides with representative pictures (**C**,**D**) (n = 5/group). The data are shown as the mean ± SE, #; *p* < 0.05 between the indicated groups using ANOVA with Tukey’s analysis.

**Figure 8 jof-09-01164-f008:**
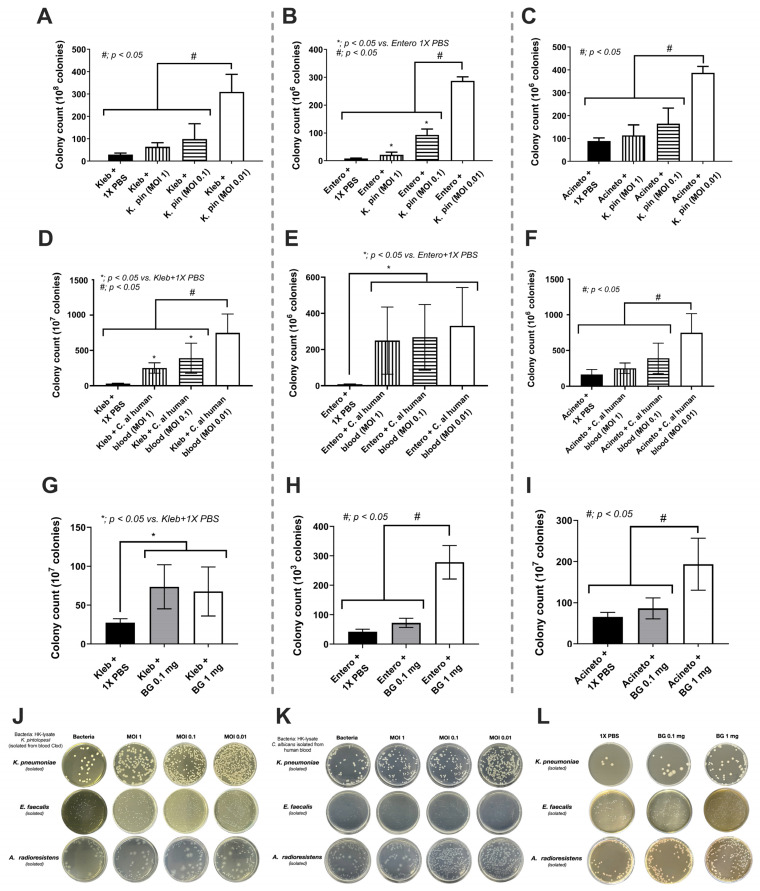
Bacterial abundance after the 24 h incubation of the isolated bacteria, including *Klebsiella pneumoniae* (Kleb), *Enterococcus faecalis* (Entero), and *Acinetobacter radioresistens* (Acineto) with the lysate from *Kazachstania pintolopesii* (K. pin) (**A**–**C**), the lysate from *Candida albicans* (C. al) (**D**–**F**), and purified (1→3)-β-d-glucan (BG) (**G**–**I**) with the representative pictures of the culture plates (**J**–**L**). The data are shown as the mean ± SE; *, *p* < 0.05 between the indicated groups and #, *p* < 0.05 between the indicated groups using ANOVA with Tukey’s analysis. Triplicate experiments were performed.

**Figure 9 jof-09-01164-f009:**
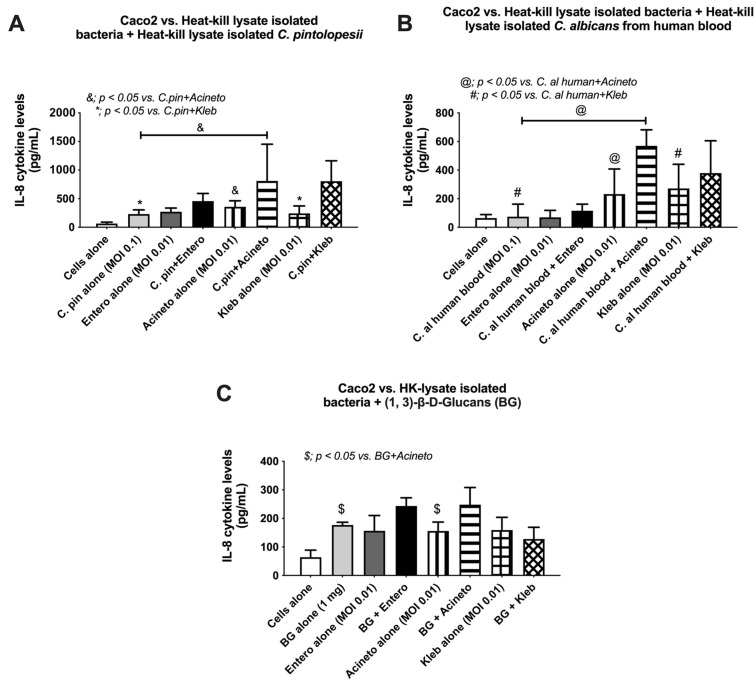
Supernatant IL-8 from enterocytes (Caco-2 cells) after 24 h incubation with the heat-kill lysate of the isolated bacteria, including *Klebsiella pneumoniae* (Kleb), *Enterococcus faecalis* (Entero), and *Acinetobacter radioresistens* (Acineto) with the lysate from *Kazachstania pintolopesii* (K. pin) (**A**), the lysate from *Candida albicans* (C. al) (**B**), and purified (1→3)-β-d-glucan (BG) (**C**) are demonstrated. The data are shown as the mean ± SE; *, *p* < 0.05 vs. K. pin+Kleb; &, *p* < 0.05 vs. K. pin+Acineto; #, *p* < 0.05 vs. C. al human+Kleb; @, *p* < 0.05 vs. C. al human+Acineto; and $, *p* < 0.05 vs. BG+Acineto using ANOVA with Tukey’s analysis. Triplicated experiments were performed.

**Figure 10 jof-09-01164-f010:**
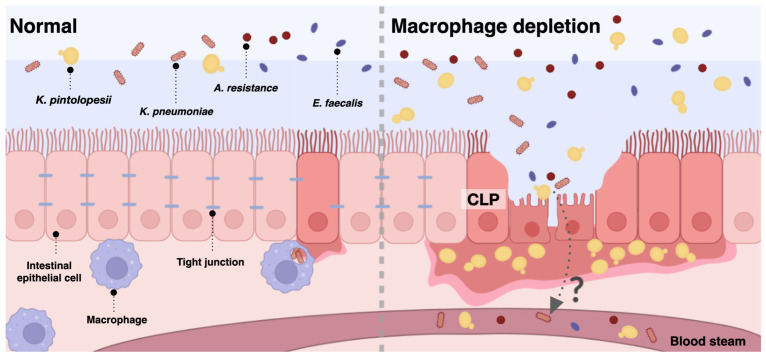
Scheme of the relationship between the normal gut microbiome and macrophage-depleted together with cecal ligation and puncture (CLP) enhancing the increases in *K. pintolopesii* and pathogenic bacteria (gut dysbiosis) and severe sepsis in a mouse model, which was generated using BioRender (https://app.biorender.com/ (accessed on 14 June 2023)).

## Data Availability

The datasets presented in this study can be found in online repositories. Mouse fecal-detected ITS1/4 sequences were deposited in the NCBI database with accession number PRJNA765503 (https://www.ncbi.nlm.nih.gov/sra/PRJNA765503 (accessed on 15 March 2023)).

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
