# Peer review of "Kazachstania pintolopesii in Blood and Intestinal Wall of Macrophage-Depleted Mice with Cecal Ligation and Puncture, the Control of Fungi by Macrophages during Sepsis"

_jof, 2023, doi:10.3390/jof9121164_

Round 1
Reviewer 1 Report
Comments and Suggestions for Authors
The author's research design based on animal models is very reasonable, and the appropriate methods can explain the key academic issues to be solved. I personally recommend accepting after major revisions. The following are the contents that need to be modified:
1. Fig 1. Where is C? Additional, it is suggested to provide a higher resolution image E (or display it in a small window magnification) to display the results of immunohistochemistry. For G, it is suggested to simultaneously display the HE staining results of consecutive sections to clearly understand the tissue structure of the image;
2. Please confirm if the resolution of Fig 3 K is consistent with other images?
3. It is suggested that Fig 4D (Group Clod CLP) should be consistent with the other three images, showing the location of the hepatic sinuses and blood vessels
Author Response
Reviewer Comments:
Reviewer 1
The author's research design based on animal models is very reasonable, and the appropriate methods can explain the key academic issues to be solved. I personally recommend accepting after major revisions. The following are the contents that need to be modified:
1. Fig 1. Where is C? Additionally, it is suggested to provide a higher resolution image E (or display it in a small window magnification) to display the results of immunohistochemistry. For G, it is suggested to simultaneously display the H&E staining results of consecutive sections to clearly understand the tissue structure of the image;
ANS: We thank the reviewer for the comment and suggestion. We rearranged the new figure as Fig. 1A-F. In addition, we provided a new Fig. 1E, as the reviewer suggested, and added the higher-resolution images. For the current Fig. 1G, we had already displayed the H&E stain of the colon in the new figure.
2. Please confirm if the resolution of Fig 3 K is consistent with other images?
ANS: We thank the reviewer for the comment. We provide the new image with a higher resolution of kidney H&E stain as in the new Fig. 3K.
3. It is suggested that Fig 4D (Group Clod CLP) should be consistent with the other three images, showing the location of the hepatic sinuses and blood vessels
ANS: We thank the reviewer for the comment. We changed new images of liver H&E staining in the group of Clod-CLP, which consists of the hepatic sinuses and blood vessels as same as the other three images.

Reviewer 2 Report
Comments and Suggestions for Authors
This manuscript is somewhat disjointed and is at times hard to follow, and not helped by the standard of English in places. To me, it feels as if this study is the amalgamation of more than one study, which makes it rather confusing in places. The key ‘take home message’ appears to be the impact of clodronate-induced macrophage depletion on the murine gut microbiota and mycobiota composition, gut dysbiosis, and gut permeability. Given the apparent negative impact of such treatment, I’m not sure why there is a need to then explore caecal puncture-induced sepsis. However, a better structured, and better explained manuscript would definitely help with the overall flow, and perhaps provide better clarity to the results presented.
In addition to this, please address the following points:
1. Incorrect species name. Candida pintolopesii is a member of the genus Kazachstania, and one of five yeast species that comprise the Kazachstania telluris species complex. There is no mention of this species complex, which I find surprising given these yeasts are opportunistic mammalian fungal pathobionts. With regard to yeast/fungal taxonomy, I strongly recommend the authors refer to the fifth edition of ‘The Yeasts: A Taxonomic Study’.
2. No details are provided regarding how the mice were housed or why male mice were specifically chosen for this study. I also assume the authors ligated the caecum 1 cm, and not 10 cm, from the caecal tip.
3. As someone who is not an immunologist and so is unfamiliar with the use of liposomal clodronate for macrophage depletion, does this drug affect any other cell types, and furthermore is the mode of action in humans the same as observed in mice? If so, then please elaborate.
4. The ITS1 and ITS 4 primers were developed as fungal-specific primers, and are not universal eukaryotic primers. Furthermore, no references are included for these primers (e.g. White et al. 1990; Gardes & Bruns, 1993).
5. The Materials and Methods section is incomplete. For instance, no details are supplied regarding culture conditions and how fungi were cultured from murine colon samples (re. Fig 6A). Furthermore, no details are provided on how fungal abundance was determined in colon tissue samples. Looking at Fig 6B, it would appear to be by qPCR, but no details are provided.
6. For the fungal community ITS profiling, were controls included, such as DNA extraction controls? Please include. In addition, why was the entire ITS region amplified? Particularly as either the ITS1- or ITS2-subregion is typically used for this type of community profiling. Indeed, given the fact that MiSeq is limited to 2x 300 bp read lengths somewhat restricts the size of full length fungal ITS amplicons that can be detected by this method. For instance, the complete ITS region (i.e. ITS1+5.8S+ITS2) of Kazachstania pintolopesii exceeds 750 bp in length, and so my question is, would this fungal species be reliably detected by this approach? This would suggest, to me at least, that this approach is perhaps flawed. Either ITS1 or ITS2 MiSeq sequencing would be better suited for gut fungal community profiling, and would perhaps provide better taxonomic resolution.
7. No details are provided regarding the number of ITS sequence reads generated, either in total, or for individual samples. This would seem an oversight, especially given that over 90% of reads in 8 of the 12 samples were classified as ‘Unidentified fungi’ (Fig 6A). Indeed in 2 Clod-CLP samples, almost all the reads appear to be ‘Unidentified fungi’.
8. Why was Candida albicans included in this study, given that its well-documented that this yeast is not a normal member of the murine gut mycobiota, as it is unable to colonise the murine GI tract without prior antibiotic pre-treatment.
9. Looking at Fig 6A to D, I do not see how the authors conclude that the increase in fecal Ascomycota may in part be due to Kazachstania pintolopesii. This yeast is a member of the Saccharomycetaceae family, which is not detected in Fig 6C. Furthermore, Kazachstania is not detected at the genus level (Fig 6D), or Candida for that matter. I would also question the conclusion that ‘the fecal fungi were mostly in the phylum Ascomycota in all experimental groups (lines 290-1), especially given that this phylum represented less than 10% of all detected fungi in many of the samples (Fig 6A). Overall, the majority of reads are classified as ‘Unidentified fungi’.
10. Given that macrophage depletion altered the composition of the gut mycobiota, does it similarly alter the gut microbiota? Do you have 16S data? This would seem an important point to discuss, given the cross-kingdom interactions that have been observed previously between bacteria and fungi.
11. The manuscript is littered with typographical and grammatical errors, which need to be corrected. Examples include ‘duirng’ (line 59), ‘amplificated’ (line 107), ‘fungal fungi’ (line 367), and ‘C. albocans’ (line 404).
12. The tone of the manuscript is, at times, somewhat vague and rather speculative, and there is a tendency to use phrases such as ‘several organisms’ and ‘some organisms’. Please be more specific and provide examples.
Comments on the Quality of English LanguageThe overall quality can be improved, and I would strongly urge the authors to seek assistance with this from a colleague whose first language is English.
Author Response
Reviewer 2
This manuscript is somewhat disjointed and is at times hard to follow, and not helped by the standard of English in places. To me, it feels as if this study is the amalgamation of more than one study, which makes it rather confusing in places. The key ‘take home message’ appears to be the impact of clodronate-induced macrophage depletion on the murine gut microbiota and mycobiota composition, gut dysbiosis, and gut permeability. Given the apparent negative impact of such treatment, I’m not sure why there is a need to then explore cecal puncture-induced sepsis. However, a better structured, and better explained manuscript would definitely help with the overall flow, and perhaps provide better clarity to the results presented.
In addition to this, please address the following points:
1. Incorrect species name Candida pintolopesii is a member of the genus Kazachstania, and one of five yeast species that comprise the K. telluris species complex. There is no mention of this species complex, which I find surprising given these yeasts are opportunistic mammalian fungal pathobionts. With regard to yeast/fungal taxonomy, I strongly recommend the authors refer to the fifth edition of ‘The Yeasts: A Taxonomic Study’.
ANS: We express our gratitude to the reviewer for this valuable comment. We had already referred to the recommended source, "The Yeasts: A Taxonomic Study 5th Edition" in its Kindle Edition, in our work as the suggestion. We added more information about C. pintolopesii as “C. pintolopesii, scientifically designated as Kazachstania pintolopesii, is an integral constituent of the genus Kazachstania and forms an essential part of the Kazachstania telluris species complex, which comprises a total of five distinct yeast species. This specific microorganism holds significant scientific relevance within the field of microbiology and contributes to the broader understanding of fungal diversity and classification”. Furthermore, we changed the word “Candida pintolopesii” to “Kazachstania pintolopesii” in the manuscript as the suggestion from the reviewer.
2. No details are provided regarding how the mice were housed or why male mice were specifically chosen for this study. I also assume the authors ligated the caecum 1 cm, and not 10 cm, from the cecal tip.
ANS: We thank the reviewer for the comment and suggestion. Because mouse gender is actually associated with the sepsis severity, only male mice were used to control this difference. We put a sentence for this aspect in the new method section. We also described the condition for housing our mouse model and edited the location of the cecal ligation at 1 cm from the cecal tip as follows “All mice were housed in a controlled-temperature environment (24 ± 2°C) with 50% relative humidity and a 12-h light-dark cycle (light from 7:00 a.m. to 7:00 p.m.). Mice were maintained with a standard diet and water during the experiment. Because mouse gender is actually associated with the sepsis severity, only male mice were used to control this difference”.
3. As someone who is not an immunologist and so is unfamiliar with the use of liposomal clodronate for macrophage depletion, does this drug affect any other cell types, and furthermore is the mode of action in humans the same as observed in mice? If so, then please elaborate.
ANS: We thank the reviewer for the comment. We have described that “For macrophage phagocytosis, the liposome characteristics, especially the size (85 nm in diameter) and the negative electrical charge, are crucial factors for achieving a specific target within the mononuclear phagocytic system (MPS) compared with dendritic cells and neutrophils. Similarly, between humans and mice, liposomal clodronate inhibits osteoclast and macrophage activity by suppressing cell proliferation. Recently, Clasteon® has been used in humans who have breast cancer with hypercalcemia. There might be different susceptibilities to macrophages among various organisms; for example, Escherichia coli, Salmonella, and Candida albicans. Indeed, the more severe sepsis in macrophage-depleted mice compared with sepsis in regular mice might be due to the enhanced leaky gut caused by gut dysbiosis with organisms that selectively survived because of macrophage depletion”.
4. The ITS1 and ITS 4 primers were developed as fungal-specific primers, and are not universal eukaryotic primers. Furthermore, no references are included for these primers (e.g., White et al. 1990; Gardes & Bruns, 1993).
ANS: We thank the reviewer for the suggestion. We change the phase from “universal eukaryotic primers” to “fungal-specific primers”. In addition, we referred to T. J. White, et al., 1990; M. GARDES and T. D. BRUNS, 1993 for the references of ITS primers.
5. The Materials and Methods section is incomplete. For instance, no details are supplied regarding culture conditions and how fungi were cultured from murine colon samples (re. Fig 7A). Furthermore, no details are provided on how fungal abundance was determined in colon tissue samples. Looking at Fig 7B, it would appear to be by qPCR, but no details are provided.
ANS: We thank the reviewer for the comment. We have already provided more details about the materials and methods in the new version of the manuscript. The fungal burden in colon tissue was performed as follows: Colon samples, each weighing 0.1 g, were gently ground until homogenized in 1xPBS (pH 7.4). Subsequently, 100 μL of each homogenized sample was spread evenly onto SDA supplemented with Chloramphenicol, and the plates were incubated at 35°C for 48 hours. Colony enumeration was performed. Furthermore, fungal abundance within the colon tissue was determined through quantitative polymerase chain reaction (qPCR). 0.1 g of colon were extracted DNA using Phenol: Chloroform methods as previously published. The fungal-specific primers, ITS1 (forward: 5’-TTCGTAGGTGAACCTGCGG-3’ and ITS4 reverse; 5’- TCCTCCGCTTATTGATATGC-3’) were used.
6. For the fungal community ITS profiling, were controls included, such as DNA extraction controls? Please include. In addition, why was the entire ITS region amplified? Particularly as either the ITS1- or ITS2-subregion is typically used for this type of community profiling. Indeed, given the fact that MiSeq is limited to 2x 300 bp read lengths somewhat restricts the size of full-length fungal ITS amplicons that can be detected by this method. For instance, the complete ITS region (i.e., ITS1+5.8S+ITS2) of Kazachstania pintolopesii exceeds 750 bp in length, and so my question is, would this fungal species be reliably detected by this approach? This would suggest, to me at least, that this approach is perhaps flawed. Either ITS1 or ITS2 MiSeq sequencing would be better suited for gut fungal community profiling, and would perhaps provide better taxonomic resolution.
ANS: We thank the reviewer for the comment. We used C. albicans ATCC90028 as a positive control in both the DNA extraction and PCR. We identified the yeast colony via ITS1 and ITS4 primers because i). primers cover the most wildly sequenced DNA region in the molecular etiology of fungi (18s rRNA to 28s rRNA region) and ii). it is commonly used in our unit. In addition, ITS2 was less powerful than ITS for resolving some closely related species (Jianping Han, et al.,2013). The subsequent species identification was conducted through sequencing, followed by the obtained sequences to a BLAST search within the NCBI GenBank database. Which, we had already described in the materials and methods section. Moreover, we would like to express our gratitude to the reviewer for their valuable suggestion regarding the potential use of ITS2 primers in future studies, which may yield more substantial insights.
7. No details are provided regarding the number of ITS sequence reads generated, either in total, or for individual samples. This would seem an oversight, especially given that over 90% of reads in 8 of the 12 samples were classified as ‘Unidentified fungi’ (Fig 6A). Indeed, in 2 Clod-CLP samples, almost all the reads appear to be ‘Unidentified fungi’.
ANS: We thank the reviewer for the suggestion. We described the number of ITS sequences as the use of DADA2 pipeline software shows a total frequency of 766,274 reads (mean 63,857 for each sample). Because of the limitation of fungal identification tools. To solve those problems in future studies we might use third-generation sequencing and change the pipelines, such as the use of nanopore sequencing. Which, we described in the discussion section.
8. Why was Candida albicans included in this study, given that its well-documented that this yeast is not a normal member of the murine gut mycobiota, as it is unable to colonise the murine GI tract without prior antibiotic pre-treatment.
ANS: We thank the reviewer for the comment. As noted by the reviewer, it is indeed recognized that C. albicans is not a typical component of the normal flora in the murine gastrointestinal tract. However, in the context of the human gut, C. albicans assumes particular significance, as it constitutes a predominant fungal species, accounting for approximately 40-60% of the fungal population. Then, to mimic the human gastrointestinal environment. We used the isolated C. albicans that were isolated from candidemia patient as gut fungi, which can translocate. This approach allowed us to create a relevant and representative experimental model mirroring the human gut microbiota, which was essential for the objectives and scope of our investigation.
9. Looking at Fig 6A to D, I do not see how the authors conclude that the increase in fecal Ascomycota may in part be due to Kazachstania pintolopesii. This yeast is a member of the Saccharomycetaceae family, which is not detected in Fig 6C. Furthermore, Kazachstania is not detected at the genus level (Fig 6D), or Candida for that matter. I would also question the conclusion that ‘the fecal fungi were mostly in the phylum Ascomycota in all experimental groups (lines 290-1), especially given that this phylum represented less than 10% of all detected fungi in many of the samples (Fig 6A). Overall, the majority of reads are classified as ‘Unidentified fungi’.
ANS: We thank the reviewer for the comment. We have already checked Fig 6A-D. Previous studies suggest that C. pintolopesii is a member of the Kazachstania (Arxiozyma) telluric species complex as the reviewer suggested above. In 1957, Kazachstania telluris was first described as Saccharomyces tellustris by van der Walt, then C. pintolopesii became a member of Saccharomyces). Which, is related to the order Saccharomycetes, the family Debaryomycetaceae, and the genus Debaryomyces (Atrayee Chattopadhyay and Mrinal K. Maiti, 2021) as in Fig 6B-D. In addition, we described a new conclusion “From fecal mycobiome analysis, there was a subtle change among the experimental groups (Fig 6A-D). As such, the fecal fungi were mostly in the phylum Ascomycota, especially in macrophage-depleted mice without sepsis (Clod-sham) and sepsis (Lipo-CLP), while macrophage-depleted CLP (Clod-CLP) and control mice (Lipo-sham) demonstrated less abundance (Fig 6E)”. Furthermore, we also added the limitation of our study in unidentified fungi as “However, our study has limitations in the application of bioinformatics tools for the analysis of unidentified fungi. Consequently, further studies are deemed essential to address these limitations comprehensively and enhance the efficacy of fungal analysis methodologies”.
10. Given that macrophage depletion altered the composition of the gut mycobiota, does it similarly alter the gut microbiota? Do you have 16S data? This would seem an important point to discuss, given the cross-kingdom interactions that have been observed previously between bacteria and fungi.
ANS: We thank the reviewer for the comment. We determined both mycobiome and bacteriome. The results showed that the pathogenic bacteria significantly increased, especially phylum Enterobacteriaceae in macrophage-depleted mice. However, the bacteriome results we described in a previous publication (Hiengrach P., et al., 2022).
11. The manuscript is littered with typographical and grammatical errors, which need to be corrected. Examples include ‘duirng’ (line 59), ‘amplificated’ (line 107), ‘fungal fungi’ (line 367), and ‘C. albocans’ (line 404).
ANS: We thank the reviewer for the suggestion. We checked and changed the typographical error to the correct words.
12. The tone of the manuscript is, at times, somewhat vague and rather speculative, and there is a tendency to use phrases such as “several organisms” and “some organisms”. Please be more specific and provide examples.
ANS: We thank the reviewer for the comment. To clearly understand, we changed “several organisms” to “both fungi and bacteria”. In addition, in the phrase “some organisms” we gave more examples referring to pathogenic bacteria (Escherichia coli and Salmonella) and fungi as the yeast C. albicans.

Round 2
Reviewer 2 Report
Comments and Suggestions for Authors
While I accept that the authors have addressed many of the points that I raised in my original review, this manuscript nevertheless still requires further improvement before it can be considered of a suitable standard for publication in JoF.
A key point that still remains to be addressed is why the entire ITS region was used for high throughput profiling of the fungal mycobiome. Whilst it is well-suited for determining species identification by colony PCR (using conventional Sanger sequencing), it is not compatible for high throughput Illumina MiSeq sequencing. Given the read length limitation imposed by MiSeq Illumina sequencing (i.e. 2x 300 bp), this is precisely why either the ITS1 or ITS sub-region is selected for fungal community profiling.
I believe this is why there is such a fundamental discrepancy between the results presented in Figure 6, where neither the Saccharomycetaceae (Family) nor Kazachstania (Genus) are detected, and the colony PCR results, where K. pintolopesii is detected. Given that K. pintolopesii was cultured from the mouse faeces, I would expect this to be reflected in the community profiling data, and it is not. This would indicate that the sequencing strategy employed in this study is flawed.
As an example, I would draw the authors attention to reference 64, where high throughput ITS1 MiSeq sequencing was used to identify K. pintolopesii as a key member of the captive cynomolgus macaque intestinal mycobiota.
In addition, the following points need to be addressed and corrected:
1. Kazachstania pintolopesii was formerly (previously) known as Candida pintolopesii. As the authors are aware, C. pintolopesii was moved to the genus Kazachstania and re-named K. pintolopesii. Therefore, this should be corrected in the Abstract.
2. Lines 44-46: I would suggest using the term ‘antibiotic-induced’ rather than ‘antibiotics-induced’.
3. Line 83-84: I would suggest using ‘temperature-controlled’ rather than ‘controlled-temperature’.
4. Lines 275-276: Please revise this sentence as it is confusingly written.
5. Line 334: Please correct ‘Debaryomyce’ to ‘Debaryomyces’.
6. Line 356: ‘Some molecules’, what does this mean? Please revise and/or be more specific.
7. Only C. albicans was used in this study, so why do the authors refer to ‘all Candida’? Please correct.
8. Given that K. pintolopesii has previously been shown to be a murine pathobiont (c/o Kurtzman and colleagues), why is there no mention to the pathogenicity of this fungus? This would seem an important oversight given the nature of this study.
9. Please refer to C. albicans and C. tropicalis in the singular and not the plural (lines 430-431).
10. Line 433: Please explain what is meant by ‘some conditions is non-surprising’.
11. Line 438: K. pintolopesii is spelt incorrectly. Please correct.
Comments on the Quality of English LanguageThe overall quality of English still needs to be improved.
Author Response
Reviewer Comments:
Reviewer 2 (Round 2)
While I accept that the authors have addressed many of the points that I raised in my original review, this manuscript nevertheless still requires further improvement before it can be considered of a suitable standard for publication in JoF.
A key point that still remains to be addressed is why the entire ITS region was used for high throughput profiling of the fungal mycobiome. Whilst it is well-suited for determining species identification by colony PCR (using conventional Sanger sequencing), it is not compatible for high throughput Illumina MiSeq sequencing. Given the read length limitation imposed by MiSeq Illumina sequencing (i.e. 2x 300 bp), this is precisely why either the ITS1 or ITS sub-region is selected for fungal community profiling.
I believe this is why there is such a fundamental discrepancy between the results presented in Figure 6, where neither the Saccharomycetaceae (Family) nor Kazachstania (Genus) are detected, and the colony PCR results, where K. pintolopesii is detected. Given that K. pintolopesii was cultured from the mouse faeces, I would expect this to be reflected in the community profiling data, and it is not. This would indicate that the sequencing strategy employed in this study is flawed.
As an example, I would draw the authors attention to reference 64, where high throughput ITS1 MiSeq sequencing was used to identify K. pintolopesii as a key member of the captive cynomolgus macaque intestinal mycobiota.
ANS: We thank the reviewer for this valuable comment and suggestion. The microbiome analysis here was performed by the Omics Sciences and Bioinformatics Center, Chulalongkorn University, a service department where we simply sent the samples to have results. We could not interfere with their use of the ITS; however, we agree with the limitations and the defects of the results that we currently have. Hence, we state this inconsistency of the results (microbiome vs. PCR) and the possible defect of the use of the current ITS as a big limitation in our manuscript. Despite the defect on ITS, the different results, at least in part, demonstrate the possible difference in fecal fungi between the macrophage-depleted group and intact macrophages which is the data that is still too low in the current literature. We will inform the correct use of ITS to the department and, hopefully, we will have a better result in our next project if we can have a grant. Here, we put this limitation in our new discussion as a technical remark in the new topic number 4.3 The discrepancy between fecal mycobiome analysis and fecal fungi from polymerase chain reaction (PCR), a possible role of the internal transcribed spacer (ITS) primers.
In addition, the following points need to be addressed and corrected:
1. Kazachstania pintolopesiiwas formerly (previously) known as Candida pintolopesii. As the authors are aware, pintolopesii was moved to the genus Kazachstania and re-named K. pintolopesii. Therefore, this should be corrected in the Abstract.
ANS: We thank the reviewer for the comment. We revised it as in the new abstract.
2. Lines 44-46: I would suggest using the term ‘antibiotic-induced’ rather than ‘antibiotics-induced’.
ANS: We thank the reviewer for the suggestion. We have already revised the phase from ‘antibiotics-induced’ to ‘antibiotic-induced’.
3. Line 83-84: I would suggest using ‘temperature-controlled’ rather than ‘controlled-temperature’.
ANS: We thank the reviewer for the suggestion. We have already revised the phase from ‘controlled-temperature’ to ‘temperature-controlled’.
4. Lines 275-276: Please revise this sentence as it is confusingly written.
ANS: We thank the reviewer for the comment. We revised the sentence in the new manuscript “The ITS sequence reads were processed using the DADA2 pipeline software, revealing a comprehensive total frequency of 766,274 reads, with an average of 63,857 reads per sample”.
5. Line 334: Please correct ‘Debaryomyce’ to ‘Debaryomyces’.
ANS: We thank the reviewer for the suggestion. We have already corrected the spelled from ‘Debaryomyce’ to ‘Debaryomyces’.
6. Line 356: ‘Some molecules’, what does this mean? Please revise and/or be more specific.
ANS: We thank the reviewer for the comment. We described ‘some molecules’ as ‘the molecules in cell wall components of K. pintolopesii’ in the new manuscript.
7. Only albicanswas used in this study, so why do the authors refer to ‘all Candida’? Please correct.
ANS: We thank the reviewer for the comment and suggestion. We changed the word from ‘all Candida’ to ‘the lysate of both isolated yeast cells (C. albicans and K. pintolopesii)’ in the result section.
8. Given that pintolopesiihas previously been shown to be a murine pathobiont (c/o Kurtzman and colleagues), why is there no mention to the pathogenicity of this fungus? This would seem an important oversight given the nature of this study.
ANS: We thank the reviewer for this valuable comment. In the new manuscript, we mentioned the pathogenicity of K. pintolopesiis as described in a previous publication “K. pintolopesiis is a fungus frequently found in the mouse intestinal mucosa and is associated with the upregulation of IL-17 receptor A (IL-17RA) and IL-23 resulting in a severe inflammatory reaction in mouse colons”.
9. Please refer to albicansand C. tropicalis in the singular and not the plural (lines 430-431).
ANS: We thank the reviewer for the suggestion. We referred to C. albicans and C. tropicalis in the singular as mentioned by the reviewer.
10. Line 433: Please explain what is meant by ‘some conditions is non-surprising’.
ANS: We thank the reviewer for the comment. We described more in the manuscript as “Because macrophage plays a crucial role in maintaining intestinal homeostasis, especially gut fungi, the fungal overgrowth in the condition without macrophages is unsurprising”.
11. Line 438: pintolopesii is spelt incorrectly. Please correct.
ANS: We thank the reviewer for the comment. We have already corrected the spelled from ‘K. pintopesii’ to ‘K. pintolopesii’.
